# CSR-Contingent Executive Compensation Incentive and Earnings Management

**Zhichuan (Frank) Li [1],\* and Caleb Thibodeau [2]**

[1]  Ivey Business School, University of Western Ontario, London, ON N6A 3K7, Canada
[2]  Department of Economics, University of Western Ontario, London, ON N6A 3K7, Canada; cthibode@uwo.ca
\*  Correspondence: Zli462@uwo.ca

**Abstract:** This paper empirically studies the connection between earnings management and corporate social performance, conditional on the existence of CSR-contingent executive compensation contracts, an emerging practice to link executive compensation to corporate social performance. We find that executives are more likely to manipulate earnings to achieve their personal compensation goals when CSR rating is low, as well as their CSR-contingent compensation. Because of public pressure on their excessive total compensation, corporate executives see no need to manipulate earnings to increase compensation when their CSR-contingent compensation is already high. Our results suggest that earnings management and CSR-contingent compensation are substitute tools to serve the interests of executives, which is an agency problem that was never previously studied. Additionally, we explore how managerial characteristics affect earnings management, driven by the incentive effects of CSR-linked compensation.

**Keywords:** earnings management; corporate social responsibility; CSR-contingent compensation; CSR contract; executive compensation; discretionary accruals

---

## 1. Introduction

Because of the recent, tremendous social and political pressure on exaggerated executive compensation, many tools that corporate top executives can use to increase their compensation changed from complement to substitute. That is, they can use one or some tools in their tool kit to achieve their personal compensation goals but refrain from using all their tools to earn excessive pay. Public pressure puts an implicit pay cap on their total annual compensation. For example, the literature (e.g., References [1,2]) discovered political and labor union pressure on CEO compensation; Mohan, Schlager, Deshpandé, and Norton [3] found that consumers avoid buying from firms with higher CEO-to-worker pay ratios. Peer pressure was also studied, for instance, for United States (US) firms versus their United Kingdom (UK) counterparts [4].

Among all the tools that managers can apply to achieve their compensation target, an emerging practice for firms to link compensation to corporate social responsibility (CSR) performance, called CSR-contingent compensation, is a new tool that managers can use to increase pay, in addition to the traditional ones such as earnings management.

Demands for regulating executive pay are regularly put forward at times of economic downturns when corporate stakeholders express stronger concerns about inequality, unfairness, and inefficiency. The financial crisis of 2008 was no exception; several governments considered or even passed laws to restrict compensation packages considered by the board or the public to be exaggerated. Meanwhile, in the last decade, corporate social responsibility (CSR) became the centerpiece of running a business, as critical sociopolitical campaigns attack corporate irresponsibility with the backing strength of social media. Corporate executives are often required by their board of directors and stakeholders to maintain

a reputable firm image, which often includes the firm's CSR rating [5]. In fact, many executives are now having their own compensation packages tied to their firms' CSR performance, an increasingly popular compensation practice called CSR-contingent executive compensation packages [6,7]. Our hand-collected data shows that about half of the Standard & Poor's (S&P) 500 companies use such CSR contracting to compensate their top executives. This type of contracting technique is not exclusive to CSR performance, and of course neither are some of the issues that arose from such contract types in the past. Traditionally, many executives' compensation is explicitly linked to corporate earnings or earnings-related performance measures. Therefore, it is not surprising that earnings management or "creative accounting" is a notably useful tool for executives wishing to artificially boost firm financial performance to achieve compensation-related benchmarks. Such measures gained notoriety in the 1990s and early 2000s with such shocking cases as WorldCom, Tyco, Xerox, Freddie Mac, and Enron. With poor intentions, there is evidence that earnings can be manipulated to help individual executives attain performance-related goals through the use of discretionary accruals [8]. These types of bonus-related compensation packages, when numerated in accounting metrics, lend to the incentive behind such types of earnings manipulation [9]. This paper empirically tests the hypotheses as to how such earnings management may manifest itself in correlation with a firm's CSR contracting when executive compensation is tied to both CSR-related benchmarks and earnings-related performance.

The use of the CSR rating in firm reputation is greatly increasing, and some view the expense paid toward increasing CSR ratings as a method of brand insurance [5]. This concept makes a great deal of sense in an age of hyper-globalization, where organization-wide branding is important to ensure positive stakeholder perception and public opinion. This could be the motivating factor behind the use of CSR-contingent compensation for executives, as boards and chairs attempt to extract the most optimal long-term profit-maximizing strategies. For a firm with a poor CSR rating, improving CSR may top the list of many invested in the company. This is important to keep in mind when we look at the relationship between the existence of social contracts with executives and the firm's CSR rating. We would expect to see executives with social contracts move their firm toward a higher CSR rating over time in order to achieve benchmarks and higher pay. While the incentive effect of such CSR contracts to improve corporate social performance is obvious and was previously studied [6], our paper finds that such compensation mitigates the earnings management problem. In particular, when managers receive rewards by achieving CSR goals, they are less likely to manage earnings to increase compensation even more, being afraid of a higher level of public scrutiny on their excessive total pay.

As our metric for earnings management, we set up a process described by Jones [10] in order to estimate the discretionary accruals of each firm for each reporting year. The absolute value of this measure gives us an idea as to what amount of earnings management is either being used to inflate or deflate financial statement values in a given period at the hand of firm executives and management. Our CSR metric for each firm is a composite rating on a scale ranging from −9 to 18, aggregating a firm's CSR strengths and weaknesses from six separate CSR categories. In addition, we use information gathered from corporate proxy statements on the presence of CSR-contingent compensation contracts and the acknowledgement of whether these contracts are objective (where dollar amount is specified in the contract) or subjective in nature.

Our analysis contributes to the design and construction of executive compensation related to firm CSR ratings. When firms design their executives' compensation contract, they should not consider each component separately. Instead, they should consider the interactions between all the components and the agency problems arising from the different incentive effects of different compositions of these components. In this paper, we study an empirical question whether self-interested and rational executives use both CSR compensation and earnings management simultaneously (a complement effect) or use them alternatively (a substitute effect) to achieve their compensation target.

Numerous studies focused on the relationship between managerial characteristics and earnings management and attempted to find what kinds of managers are more likely to manage earnings. In this paper, we provide a unique experiment lab, the conditional existence of CSR-contingent compensation

incentive, to study the effect of executive age and gender on earnings management. We contribute to the literature by showing that, with different incentives in place, managers, even with same characteristics, may exhibit completely different preference and activities.

## 2. Literature Review and Discussion

Agency theory is at the heart of the issue explored in this paper, which is an economic theory describing a deviation in incentives between the principal stakeholders of a company and the agents or management at the helm. Bebchuck and Fried [11] studied the relationship between executive compensation and the agency problem, and suggested that the design of the compensation itself can be a portion of the agency puzzle itself. Because many activities of the management are unobservable by or hard to evaluate by the board or the shareholders and stakeholders of a firm, the firm has to link executive compensation to some simple, performance-related measures. These measures or targets, although they have strong incentive effects, can be easily manipulated by executives for their personal benefits. Prior, Surroca, and Tribo [12] argued that this can be a downfall of the CSR rating, since a firm's positive social image can be used to provide a blanket for executives to shield scrutiny or negative news. They found that there is a positive correlation between levels of earnings management and the CSR rating. Our paper suggests that this relationship depends on executive incentives. To avoid executives using CSR rating as a safety net for their misbehavior, the boards either need to monitor the executives more closely or provide proper incentives to compensate them for achieving the company goals. Executives respond to these incentives by working harder and better and refraining from unethical activities.

CSR can plausibly impact a firm's corporate financial performance (CFP). The research on the relationship between CSR and CFP is saturated (e.g., References [13,14]). Many studies found that CSR can improve CFP through a variety of channels, such as labor reputation [15], customer awareness [16], labor productivity [17], and improved transparency [18]. Some studies found a negative relationship (e.g., Reference [19]), citing an agency problem or shareholder–stakeholder conflicts. Many studies found no significant relationship between CSR and firm value (e.g., Reference [20]). Margolis, Elfenbein, and Walsh [21] conducted a meta-analysis of many such empirical studies and concluded that the relationship between CSR and firm value is positive but small. A recent paper by Hong, Li, and Minor [22] found evidence that CSR, which can be improved by CSR-contingent executive compensation, is in fact beneficial to shareholders as opposed to an agency cost.

The literature extensively studied the relationship between executive compensation and earnings management (e.g., References [23–26]). All these prior studies indicated that managers engage in earnings management to maximize their total pay because most firms have earnings-based executive compensation contracts, explicitly or implicitly. A closely related paper to ours is that by Shuto [27], which discovered that, in Japanese firms, earnings management increases executive compensation, and managers who receive no bonus are more likely to manipulate it to increase their future bonus.

Our first formalized query is to ask what effect CSR ratings have on levels of earnings management at firms with CSR-contracted executives versus those without. We attempt to discover here if there is a larger tendency toward managing earnings when parts of executive compensation are "on the line", i.e., when CSR performance measures and CSR compensation targets are not being fully achieved. We expect that executives may be more uncomfortable and greedier with lower CSR ratings in the case of their compensation being directly impacted by CSR. Thus, we look for an indication of misbehavior and whether or not earnings management subsides as the CSR rating of a firm gets higher. This notion is in contrast to Prior et al. [12], who found more earnings management under the cover of higher CSR ratings.

More generally, we take a look at whether firms with executives under social contracts exert a higher level of earnings management than those firms without social contracts. We attempt to find whether CSR-linked compensation incentive plays an important role in mediating the relationship between earnings management and CSR ratings. In particular, we expect the relationship between CSR

rating and earnings management to be more significant when executives have a CSR contract. To attain personal compensation goals, when the CSR rating is low, as well as CSR-contingent compensation, they use earnings-related compensation to substitute their CSR-related compensation. Therefore, we should observe that executives with CSR contracts are more likely to manipulate earnings when CSR rating is low.

Our second goal is to explore the effect of managerial characteristics, specifically age and gender, on earnings management. Age is an important determinant of earnings management because of the horizon problem for CEOs nearing retirement age. The literature (e.g., Reference [28]) discovered that CEOs nearing the retirement age are associated with aggressive income-increasing earnings management. However, other researchers found that older managers are less likely to manage earnings (e.g., Reference [29]).

While prior research provided some insights into the role of gender on earnings management, the literature is nascent and the reported results are mixed. For example, Peni and Vahamaa [30] and Barua, Davidson, Rama, and Thiruvadi [31] found that female Chief Financial Officers (CFOs) are associated with less earnings management, whereas female CEOs are not. Yet, Lakhal, Aguir, Lakhal, and Malek [32] concluded that more female directors reduce earnings management, but that female CEOs and female CFOs do not affect earnings management. Finally, Ye, Zhang, and Rezaee [33] offered evidence that gender does not influence earnings management. These earlier studies, however, failed to consider the role of compensation incentive as a mediator of the relationship between managerial characteristics and earnings management. Our paper contributes to the literature by providing an important mediator and by explaining why prior studies reported conflicting results.

## 3. Methods and Data

### 3.1. Datasets

The datasets used in the analysis originated from three different sources.

Compustat: The WRDS directory was used to retrieve annually reported data on firm variables; executive characteristics were retrieved from the Execucomp database beginning in the year 1991 until the most recently available year of 2017. The variables included the following:

- Total current assets (CA);
- Total current liabilities (CL);
- Cash (Cash);
- Long-term debt due in one year (DD1);
- Depreciation and amortization expenses (Dep);
- Net sales/turnover (SALE);
- Gross property, plant, and equipment total (PPE);
- Total assets (AT);
- Standard industrial classification (SIC) codes;
- Executive age;
- Executive gender.

MSCI: The WRDS directory was used to retrieve annually reported data on firm CSR ratings based on the six categories listed below. These categories contain the reported strengths and concerns in each, and aggregate to the overall CSR rating. Observations from the database began in the year 1995 until the most recently available year of 2013. The variables were as follows:

- Community;
- Diversity;
- Employee relations;
- Environmental;

- Human rights;
- Product.

Unlisted: Please see details under the Section "Data and Variable Description" of Ikram et al. [7]. The dataset included information about the following:

- Whether an executive has CSR-related compensation contract (binary);
- Objectivity of the CSR contract (binary).

In particular, a CSR-contingent compensation contract includes a compensation component directly linked to all the keywords related to the following categories:

- Community;
- Ethic;
- Satisfaction;
- Environment;
- Sustainability;
- Safety;
- Health;
- Injury;
- Accident;
- Diversity.

The sample period started from the year 1991 because of the scarcity of CSR-related data predating 1992, where 1991 was used to draw the lagged total assets variable. The CSR index gained more prominence and popularity lately, with a cross-industry average of 72% reporting out of 4100 companies in the year 2013, as stated by a report published by KPMG. The early CSR data on MSCI in the early 1990s are far less complete. It is for this reason that our results tend to be more weighted toward recent years and, therefore, reflect more recent CSR trends.

### 3.2. Methodology

The first metric required for all the sample firms was earnings management, the input dataset of which was retrieved from the Compustat library. We applied the discretionary accrual approach outlined in Jones [10] and used in Reference [34] to calculate this measure.

Firstly, the data collected on firm assets, liabilities, cash, debt, and sales were lagged for one period in order to create year-over-year changes or differences, denoted by $\Delta$. The description of the total accruals for firm $i$ at year $t$ is given by

$$TACC_{i,t} = \frac{\left(\Delta CA_{i,t} - \Delta CL_{i,t}\right) - \left(\Delta Cash_{i,t} - \Delta DD1_{i,t}\right) - Dep_{i,t}}{AT_{i,\,t-1}} \tag{1}$$

The total accruals were calculated for the initial sample of total 144,696 observations. Note that the total accruals for each firm in each observed year are normalized by the lagged assets of said firm.

Next, we removed the non-discretionary accruals from this total accrual measure, in order to identify the portion of earnings management that can be attributed to executives, as in Dechow et al. [34]. We started by estimating the following regression across all industry sectors:

$$TACC_{i,t} = \alpha_0 + \alpha_1\left(\frac{1}{AT_{i,t-1}}\right) + \alpha_2\frac{\Delta Rev_{i,t}}{AT_{i,t-1}} + \alpha_3\frac{PPE_{i,t}}{AT_{i,t-1}} + \varepsilon_{i,t} \tag{2}$$

We controlled for industry classifications based on the standard industrial classification (SIC) codes. The change in sales (Rev), and in gross property, plant, and equipment (PPE) were normalized by the lagged assets. We used alternative firm size measures according to Dang, Li, and Yang [35] and found qualitatively similar results.

The non-discretionary accruals of each firm were estimated based on the following model as in Jones [10]:

$$\overline{NDACC}_{i,t} = \overline{\alpha_0} + \overline{\alpha_1}\left(\frac{1}{AT_{i,t-1}}\right) + \overline{\alpha_2}\frac{\Delta Rev_{i,t}}{AT_{i,t-1}} + \overline{\alpha_3}\frac{PPE_{i,t}}{AT_{i,t-1}} + \varepsilon_{i,t}. \tag{3}$$

Subsequently, we used our formulation of the total accruals from Equation (1) to subtract Equation (3) to arrive at our estimate of the total discretionary accruals in Equation (4).

$$\overline{DACC}_{i,t} = TA_{i,t} - \overline{NDACC}_{i,t} \tag{4}$$

For the formulation of CSR ratings, data extracted under six main headings were aggregated as follows:

$$
\begin{aligned}
CSR\ rating_{i,t} = \ & (community\ strengths_{i,t} - community\ concerns_{i,t}) \\
& + (diversity\ strengths_{i,t} - diversity\ concerns_{i,t}) \\
& + (employee\ relation\ strengths_{i,t} \\
& - employee\ relation\ concerns_{i,t}) \\
& + (environmental\ strengths_{i,t} - environmental\ concerns_{i,t}) \\
& + (human\ rights\ strengths_{i,t} - human\ rights\ concerns_{i,t}) \\
& + (product\ strengths_{i,t} - product\ concerns_{i,t}).
\end{aligned}
\tag{5}
$$

The CSR ratings in the data were available for the period of 1995–2013. There were 28,781 observations that supplied CSR ratings over this time period, ranging from a score of −9 to 18 on the ratings scale with the highest (in numerical value) score being the best.

The data of CSR-contingent contracts included firms that explicitly stated in their proxy statements that such contracts exist. The CSR contract dummy variable was equal to 1 in the case of such a contract and 0 if no such contract exists in a firm. Furthermore, we defined a CSR-contingent contract as "objective" if the executive receiving the contract knew ex ante how much he/she could expect to earn from pursuing pre-specified CSR-related activities. An objective CSR-contingent compensation was deemed "formulaic" if the contract specified the weights attached to the accomplishment of specific CSR-related activities. Conversely, we defined a CSR contract as "subjective" if the executive receiving the contract was ex ante unaware of how much he/she could expect to earn from engaging in specific CSR activities. That is, the percentage or amount of reward was ex ante unknown to the executive and subject to the discretion of the company ex post. The dummy variable of objective contract was equal to 1 if the CSR contract was objective, and 0 if subjective.

## 3.3. Regressions

We used the below set of regression specifications to identify the relationships between earnings management, CSR rating, and CSR contract. We hypothesize that there is a link in a firm between the amount of earnings management taking place and CSR rating during the same time period, conditional on the existence of a social contract. Our regression analysis follows in four segments, namely discretionary accruals with CSR-contingent contracts and CSR rating, discretionary accruals with age and gender, discretionary accruals with CSR rating and contract objectivity, and contract objectivity with age and gender.

Our first model attempted to investigate whether executives at firms with social contracts were more likely to manage earnings, in order to shape firm earnings or CSR performance more conducive in achieving their compensation targets. The model had the capacity to show whether CSR rating had a larger effect on earnings management at firms with social contracts. The use of the interaction term was most important in this analysis, relying on a binary "CSR contract" variable.

$$\left|DACC_{i,t}\right| = \beta_0 + \beta_1\ CSR\ rating_{i,t} + \beta_2\ CSR\ contract_{i,t} + \beta_3(CSR\ rating_{i,t} * CSR\ contract_{i,t}) + \varepsilon_{i,t} \tag{6}$$

Our second model took a more simplistic approach to study whether CSR rating and CSR contract objectivity had any explanatory power for earnings management.

$$\left|DACC_{i,t}\right| = \beta_0 + \beta_1 CSR\ rating_{i,t} + \beta_2 Objective\ CSR\ contract_{i,t} + \varepsilon_{i,t} \tag{7}$$

Our third model focused on executive age and gender to examine whether these managerial characteristics were linked to earnings management, conditional on the existence of CSR contracts. This may lead to indications about what kinds of managers have a higher propensity to use earnings management with CSR-contingent compensation incentives. Specifically, we examined the effect of age and gender and their interactive effect on earnings management, conditional on the existence of CSR contracts. Gender was equal to 1 if male and 0 if female.

$$\begin{aligned}\left|DACC_{i,t}\right| = {} & \beta_0 + \beta_1\ Exec\ Age_{i,t} + \beta_2\ Exec\ Gender_{i,t} \\ & + \beta_3\left(Exec\ Age_{i,t} * Exec\ Gender_{i,t}\right) + \varepsilon_{i,t}\end{aligned} \tag{8}$$

Our last model took a simple approach to explore whether the objectivity of a social contract can be explained by the age and sex profile of the executive in question. There may be an executive type which is more exposed to accepting or participating in a CSR-contingent compensation contract or to being involved with a notably objective contract.

$$Objective\ CSR\ contract_{i,t} = \beta_0 + \beta_1 Exec\ Age_{i,t} + \beta_2 Exec\ Gender_{i,t} + \varepsilon_{i,t} \tag{9}$$

To mitigate the omitted variable bias, we controlled for firm fixed effects to control for unobservable time-invariant firm and industry characteristics. For robustness checks, we also controlled for lagged dependent variables in untabulated results to alleviate the endogeneity problem [36].

## 4. Results

Based on our first model, which connects the firms with and without social contracts to their propensity to manage earnings under the effect of their CSR ratings, we obtained the results in Table 1.

**Table 1.** CSR rating and CSR contract and their interactive effect on earnings management.

| Variable | Estimate | Standard Error | *T*-Value | Pr > |t| |
|---|---|---|---|---|
| $\beta_1$ | 0.0023850411 | 0.00426659 | 0.56 | 0.5762 |
| $\beta_2$ | −0.0253712011 | 0.02077070 | −1.22 | 0.2219 |
| $\beta_3$ | −0.0267204754 | 0.00570583 | −4.68 | <0.0001 |
| $R^2$ | 0.115826 | | | |
| \|DACC\| median | 0.0441969 | | | |
| \|DACC\| mean | 0.119147 | | | |

Table 1 provides three implications. Firstly, when there exists a CSR contract (CSR contract dummy = 1), the aggregate effect of CSR rating on earnings management ($\beta_1 + \beta_3 = -0.024$) is negative and significant. As firm CSR ratings increase, we see that earnings management decreases at firms with social contracts in place. Secondly, we see little to no effect of the CSR rating on earnings management at firms with no social contracts ($\beta_1 = 0.002$ and insignificant). Without the CSR compensation incentive, CSR rating is not related to earnings management. No matter how high or low the CSR rating is, it does not affect executive compensation and, therefore, does not affect the tendency for executives to manage earnings. Thirdly, $\beta_2$ and $\beta_3$ are both negative, suggesting that the aggregate effect of CSR contract on earnings management is negative. CSR contracting is a useful method to deter earnings management. Overall, these results suggest that, when executives can receive rewards through their CSR contracts by improving CSR rating, they are less likely to manipulate earnings; otherwise, their excessive compensation will be under public scrutiny.

From our second model, which looks at the descriptors of CSR rating and the objectivity of a social contract, we obtained the results in Table 2.

**Table 2.** CSR rating, objective CSR contract, and earnings management.

| Variable | Estimate | Standard Error | T-Value | Pr > \|t\| |
|---|---|---|---|---|
| $\beta_1$ | −0.0072244431 | 0.00383714 | −1.88 | 0.0598 |
| $\beta_2$ | 0.0278914879 | 0.01682793 | 1.66 | 0.0975 |
| $R^2$ | 0.113355 | | | |
| \|DACC\| *median* | 0.0441969 | | | |
| \|DACC\| *mean* | 0.119147 | | | |

A negative $\beta_1$ suggests that CSR rating and earnings management are negatively correlated; a positive $\beta_2$ indicates that an objective CSR contract increases the level of discretionary accruals. Plausibly, when managers do the math to calculate their CSR-related compensation based on the objective contracts, they are more likely to manipulate earnings to achieve their compensation goals more precisely. In conjunction with the previous result, this implies that executives with social contracts are more likely to manage earnings and typically will do so when the firm CSR rating is lower or when the CSR contract is objective. We do not see such behaviors in executives who do not have a social contract, which reinforces the idea that it is executive incentives which impact earnings management.

Now, turning to personal characteristics of executives undertaking earnings management and having CSR contracts, our third model obtained the results in Table 3.

**Table 3.** Executive characteristics and earnings management.

| Variable | Estimate | Standard Error | T-Value | Pr > \|t\| |
|---|---|---|---|---|
| $\beta_1$ | −0.0057685005 | 0.00240172 | −2.40 | 0.0163 |
| $\beta_2$ | −0.2436452116 | 0.12357294 | −1.97 | 0.0487 |
| $\beta_3$ | 0.0044890732 | 0.00244927 | 1.83 | 0.0668 |
| $R^2$ | 0.157697 | | | |
| \|DACC\| *median* | 0.0441969 | | | |
| \|DACC\| *mean* | 0.119147 | | | |

For a female executive (gender dummy = 0), the relationship between age and earnings management is $\beta_1$. The level of earnings management decreases with age, which is inconsistent with the literature which typically found a positive relationship [28] and suggested that managers, nearing the retirement age, have a short horizon and, therefore, engage in more aggressive income-increasing earnings management. We found that this horizon problem may be more serious for male executives. Taking $\beta_3$ into account, the negative relationship ($\beta_1 + \beta_3$), which is close to zero and insignificant, is much weaker for males. Surprisingly, a significantly negative $\beta_2$ suggests that, on average, male executives are less likely to participate in earnings management. In sum, while males have more of the age/horizon problem, they are on average less likely to manage earnings.

Our results in Table 3, in contrast to the literature, suggest that the conclusions in the current literature may be spurious due to missing an important consideration: compensation incentives. Our unique setting of CSR contract provides a useful experiment lab, where we observe that the relationships between managerial characteristics and earnings management may indeed be driven by the incentive effects of CSR compensation. With different incentives in place (i.e., with CSR contract versus without), managers, even with same characteristics, may exhibit completely different preference and activities.

The last model shows how executive age and gender match to objective CSR contracts (Please see Table 4).

**Table 4.** Executive characteristics and objective CSR contracts.

| Variable | Estimate | Standard Error | *T*-Value | Pr > \|t\| |
|----------|----------|----------------|-----------|------------|
| $\beta_1$ | 0.0056548035 | 0.00065936 | 8.58 | <0.0001 |
| $\beta_2$ | 0.0248645451 | 0.01909682 | 1.30 | 0.1929 |
| $R^2$ | 0.029630 | | | |

$\beta_1$ is positive, implying that older executives are more likely to receive objective CSR contracts, but this magnitude coefficient has a relatively minute effect. Gender has no significant effect on receiving objective or subjective contracts.

The two tables below are derived from the datasets used for the regression analysis. They supply several further clues as to the behavior observed in the results based on the above multivariate analysis. Indeed, causality is not a consequence of correlation, but these correlations help expand the connection between parameters and variables.

Table 5 shows that CSR rating is negatively correlated with CSR contract and objective CSR contract. One explanation is that firms with currently low CSR ratings are more likely to use CSR contracting in order to improve future CSR performance. Additionally, because objective CSR contracts are more likely to be effective due to their stronger incentive nature (i.e., a specific reward amount is prespecified in such contracts), low-CSR firms are more likely to use objective contracts.

**Table 5.** Correlations among earnings management, CSR rating, and CSR contract.

| Correlation | \|*DACC*\| | CSR Rating | Contract Strength | Contract (Y/N) |
|-------------|-----------|------------|-------------------|----------------|
| \|*DACC*\| | 1 | | | |
| CSR Rating | −0.03369 / 0.0032 | 1 | | |
| Contract Strength | 0.03138 / 0.0060 | −0.07440 / <0.0001 | 1 | |
| Contract (Y/N) | 0.02523 / 0.0272 | −0.09604 / <0.0001 | 0.93158 / <0.0001 | 1 |

One major point to observe from these correlations in Table 6 is that executive age and gender do not have a very significant correlation with earnings management, but they do have a very strong correlation with CSR rating. In the second column, CSR rating has a strong relationship with female executives and age. We cannot draw causality in this correlation analysis. These relationships can be driven by firm-manager assertive matching; older and/or female executives may be hired by firms with a high CSR rating.

**Table 6.** Correlations among earnings management, CSR rating, and executive age and gender.

| Correlation | \|*DACC*\| | CSR Rating | Exec Age | Exec Gender |
|-------------|-----------|------------|----------|-------------|
| \|*DACC*\| | 1 | | | |
| CSR Rating | −0.00705 / 0.0366 | 1 | | |
| Exec Age | −0.00099 / 0.8104 | 0.01926 / <0.0001 | 1 | |
| Exec Gender | −0.00624 / 0.0644 | −0.06969 / <0.0001 | 0.09632 / <0.0001 | 1 |

## 5. Conclusions

In this paper, we embarked to discover if any links exist between the use of CSR-contingent compensation contracts and the use of earnings management to feign firm performance. Alongside this analysis, some basic and well-studied executive characteristics were taken into account, as well as the firm CSR rating.

Our first finding revealed the connection between the level of earnings management and CSR rating at a firm, conditional on the existence of CSR-contingency compensation. Our results suggest that CSR rating is negatively correlated with earnings management, and this relationship is solely driven by the executives who receive CSR-contingent compensation incentives. Because of this CSR-linked compensation, CSR rating increases executives' compensation and, therefore, decreases the need to manipulate earnings. The relationship between CSR rating and earnings management was not significant for firms without CSR compensation, simply because CSR rating does not affect executive compensation in these firms and, therefore, does not affect executives' incentive to manage earnings. This conclusion makes more sense in the context of the increasingly strong pressure on executive total pay which implicitly puts a cap (target) on executives' total compensation level.

This evidence is in contrast to the work of Prior et al. [12] and Salewski and Zülch [37], who argued that there exists a positive relationship between earnings management activity and CSR ratings because managers use CSR ratings to appease stakeholders and to create a safety net for their earnings management or other misbehaviors. Our results imply that CSR-related compensation is an effective tool to mitigate this agency problem in the literature. Such compensation contracts not only explicitly encourage executives to improve social performance of their firms, but also implicitly reduce their incentive to manage earnings to jack up their total pay. The second benefit may be more important than the first one for shareholders who usually rely on earnings information to evaluate firms and allocate their investments.

Our findings based on executive characteristics contribute to the literature by showing that these characteristics do not have a static impact on earnings management. Instead, managers with different characteristics respond to incentives differently and, therefore, without controlling for executive incentives, the research on personal characteristics will be spurious and inconclusive.

One limitation of this paper is that we did not explore the effects of corporate governance in the context of CSR contracting. For the future research, it is potentially important to study the mediating effects of corporate governance measures, such as market competition [38], mutual monitoring among executives [39,40], inside debt [41], and CEO tournament [42]. Studying these aspects of corporate governance is one step toward opening the black box of the firm effects on mitigating the agency problem we discovered in this paper. Another direction for future research is to consider more managerial characteristics, especially the proxies for managerial power [43] and the unobservable managerial ability and risk aversion [44], all of which can possibly affect the executives' decision on earnings management and CSR initiatives. Finally, researchers can test our hypotheses in the international context, especially in countries with less public pressure on executive total compensation [45].

**Author Contributions:** Conceptualization, Z.(F.)L.; methodology, C.T.; software, C.T.; validation, Z.(F.)L. and C.T.; formal analysis, Z.(F.)L. and C.T.; investigation, Z.(F.)L. and C.T.; resources, Z.(F.)L. and C.T.; data curation, Z.(F.)L. and C.T.; writing—original draft preparation, C.T.; writing—review and editing, Z.(F.)L.; supervision, Z.(F.)L.

**Funding:** This research received no external funding.

**Conflicts of Interest:** The authors declare no conflicts of interest.

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
