# Peer review of "CSR-Contingent Executive Compensation Incentive and Earnings Management"

_sustainability, doi:10.3390/su11123421_

Round 1
Reviewer 1 Report
The main goal of the manuscript was to study the relationship between the use of earnings management and CSR ratings for executives with social contracts.
The research area is interesting and so is the posed question. However, the biggest shortcoming of the paper is the lack of theoretical modelling preceding the statistical studies.
In order to present hypothesis/ research questions and then study regression – you need to have a strong theoretical/ empirical justification to do so, which is lacking in your manuscript.
In your Introduction and Literature review you do not give enough theoretical background and you do not develop a theory justifying why you would look for connections between earnings management and CSR ratings, earnings management and executive’s age/ gender and CSR ratings and executive’s age/ gender. Is there any previous research indicating that gender/ age matters in case of earnings management or CSR ratings? Is there a reason why it should matter? What is the theoretical and empirically tested connection between earnings management and CSR ratings in previous studies? What is the mechanism that would lead to correlation between the two?
Due to very modest literature review and theoretical modelling – I find your research limited in scientific soundness. I would recommend a more thorough literature study concentrating of a specific matter. I think you are trying to explore too many areas without developing a theoretical framework first. Lack of theoretical justification leaves a reader with a sense that some of the tests are done randomly (i.e. age or gender connection). It is interesting in the exploratory stage of the research, but just testing correlations without a preceding theoretical justification is not enough, in my opinion, to develop sound conclusions. Please, develop a more thorough literature review, indicate the research gap and your contribution. When presenting a research design – please add theoretical justification and adequate referencing to every relationship you are studying. If your main interest is CSR ratings vs. earnings management – you need to present the conclusions of a very wide research on CSR and corporate financial performance, to explain the possible influence of executive social contracts.
Also, it would be helpful if you defined what you mean by social/CSR related contracts. What benchmarks do they typically include that could (and how?) influence executives’ behaviour in earnings management? The hypothesized connection between the two is most unclear to me. How would earnings management lead to improvement of CSR ratings and hence – to the executives’ gain? Did you check for other variables? Maybe companies using social contracts include also financial benchmarks to incentivise their managers. In such a case a correlation you are getting might be related to a different variable.
I think you found an interesting research topic. Your regression model is showing that there is a case to study there, however, you need to present a better understanding of the theoretical context that would justify your model and conclusions.
Also, please correct the referencing to match the journal format.
Author Response
Thanks so much for all the insightful and specific comments which helped us improve the paper considerably.

Reviewer 2 Report
The paper under review refers to an interesting and current topic, then providing interesting findings from the application of the suggested model. However, some changes should be done prior to considering its publication. To be precise:
a) In a general sense, a review by an English native speaker or expert would be advisable. Maybe the quality of writing is quite good from a grammar view, but there are a real number of spelling mistakes in words all along the text.
b) Talking on the proper content, there is a scarce use of bibliographical references in both the introductory section and the second one on literature review and discussion. Moreover, such references are not so up to date, as a number of them are dated more than ten years ago (namely in 90's) which is not so consistent with the current character of the topic. These two sections should be reconsidered (and subsequently rewritten) on the basis of a major number of current bibliographical references.
c) What is more, even if referring to those bibliographical sources which are currently used, there are some mistakes which should be amended. So:
- When naming the authors of a paper contribution as part of the sentences, the names should not be included in parenthesis together with the date of the contribution, but only the year. Fr example, in page 2 (line 54) we can read "…we set up a process described by (Jones, 1991) in order to…" which should be turned into "…we set up a process described by Jones (1991) in order to…". There are a number of similar situations.
- The paper by Bergstresser and Philippon in 2006 is quoted in pages 1 and 3 as authored by Bergstressa and Philippon in 2005. Both the name of the first co-author and the date should be properly indicated.
- There is a footnote in page 1 providing a link to the site of The Harvard Business Review. If clicking the link, you can see this is a paper by Jen Boynton in 2013. So, the quotation should be to Boynton (2013) and properly included in text. This reference should also appear in the final list of bibliographical references.
- A similar situation appears regarding the footnote in page 2, as the link to the MIT site is in fact a link to the Lecture Note on Agency Theory by Robert Gibbons which should be quoted on the name of the author as well as included in the final list of bibliographical references.
- Then, the contribution by Vartiak (2016) which is mentioned in page 4 (lines 144-145) is not included in the final list of bibliographical references and it should be, moreover when taking in mind the apparent scarce use of updated references which was above mentioned. A similar situation occurs with the contribution by Li et al. (2017) which is mentioned in page 9 (line 335).
- Including new bibliographical quotations in the section on conclusions is not recommended, namely when such contributions have not been previously quoted in previous sections of the text. In this case, we can see quotations to the references by Li et al. (2017) in page 9 or by Prior et al. (2007) in page 10, but there is an additional quotation of a contribution by Salewski and Zülch in 2014 which is mentioned by the first time, this being not so adequate. Moreover, if having a look at the final list of references, the information provided on this contribution is incomplete, as there is only a title and we cannot guess if it is a book, a chapter, a paper or whatever.
- Finally, a contribution by Ikram, Li and Minor (2017) is included in the list of bibliographical references even when it is not previously quoted at any place along the text. So, it should be removed unless at the end it is the above mentioned 'missed' reference to the contribution by Li et al. (2017) in page 9. At any case, and similarly to Salewski and Zülch (2014) only a title is included and the complete information on the contribution should be provided.
Author Response

(The authors gave the same response as above.)

Reviewer 3 Report
Dear Authors,
1. The aim of the paper should be defined more precisely in the Introduction section.
2. Methodology should be shortly mentioned in the Abstract.
4. The Literature Review section should be significantly developed (suggest at least doubling the number of references). See for example:
- Muhammad Safdar Sial, Zheng Chunmei, Tehmina Khan, Vinh Khuong Nguyen,(2018) "Corporate social responsibility, firm performance and the moderating effect of earnings management in Chinese firms", Asia-Pacific Journal of Business Administration, Vol. 10 Issue: 2/3, pp.184-199, https:// doi.org/10.1108/APJBA-03-2018-0051
5. Rethink the “Conclusions” section - at the moment it lacks the Limitations and the Future Research subsections. Try to:
·highlight key findings in your “Results” section,
·place the paper within the context of how your research advances past research about the topic,
·describe how a previously identified gap in the literature (your literature review section) has been filled by your research,
·demonstrate the importance of your ideas and recommendations/suggestions,
·define the limitations of your research,
·propose possible new or expanded ways of thinking about the research problem.
I do hope you find the comments helpful as you move forward with your paper.
Author Response

(The authors gave the same response as above.)

Round 2
Reviewer 2 Report
The paper under review has been clearly improved by the authors on the basis of the provided comments and suggestions. Thus, it may be considered for publication in this new version.
Reviewer 3 Report
Dear Authors,
Thank you very much for this revision. I accept the article in the present form.
Good luck with your future resaerch!